# Analytical Model of CVD Growth of Graphene on Cu(111) Surface

**DOI:** 10.3390/nano12172963

**Published:** 2022-08-27

**Authors:** Ilya Popov, Patrick Bügel, Mariana Kozlowska, Karin Fink, Felix Studt, Dmitry I. Sharapa

**Affiliations:** 1Institute of Catalysis Research and Technology (IKFT), Karlsruhe Institute of Technology (KIT), Hermann-von-Helmholtz-Platz 1, 76344 Eggenstein-Leopoldshafen, Germany; 2School of Chemistry, University of Nottingham, University Park, Nottingham NG7 2RD, UK; 3Institute of Nanotechnology (INT), Karlsruhe Institute of Technology (KIT), Hermann-von-Helmholtz-Platz 1, 76344 Eggenstein-Leopoldshafen, Germany; 4Institute for Chemical Technology and Polymer Chemistry (ITCP), Karlsruhe Institute of Technology (KIT), 76131 Karlsruhe, Germany

**Keywords:** graphene growth, nucleation kinetics, analytical model, chemical vapor deposition, lattice gas model

## Abstract

Although the CVD synthesis of graphene on Cu(111) is an industrial process of outstanding importance, its theoretical description and modeling are hampered by its multiscale nature and the large number of elementary reactions involved. In this work, we propose an analytical model of graphene nucleation and growth on Cu(111) surfaces based on the combination of kinetic nucleation theory and the DFT simulations of elementary steps. In the framework of the proposed model, the mechanism of graphene nucleation is analyzed with particular emphasis on the roles played by the two main feeding species, C and $C_{2}$. Our analysis reveals unexpected patterns of graphene growth, not typical for classical nucleation theories. In addition, we show that the proposed theory allows for the reproduction of the experimentally observed characteristics of polycrystalline graphene samples in the most computationally efficient way.

## 1. Introduction

Since its discovery in 2004 [1], graphene has become a material with a wide range of possible scientific and industrial applications [2,3,4,5]. With the increase in graphene consumption, multiple methods of its production have been suggested, including exfoliation [6], the hydrothermal reduction of graphene oxide [7], carbon dioxide reduction [8], and chemical vapor deposition (CVD) [9] and its plasma-enhanced version (PECVD) [10]. For a comparative overview of different techniques, we refer to the topic-focused Refs. [11,12,13,14].

Among the mentioned methods, CVD is most frequently used for the production of high-quality graphene having high homogeneity, imperviousness, high purity, and fine grains. It is used nowadays to produce hundreds of thousands of square meters of graphene annually [15].

This resulted in an intensive study of the CVD graphene growth process, which showed that on different surfaces/substrates and in different conditions qualitatively different graphene samples may be obtained. For instance, while on Ru, Ni, Ir, and Co surfaces, the segregation growth mechanism takes place, typically yielding bi- or multilayer graphene [16]. The growth on Cu (which has a low solubility of carbon) is governed exclusively by surface processes, which allows for the synthesis of monolayer samples. While graphene formed on the Cu(001) surface is polycrystalline and not uniform, Cu(111) allows for the production of single-layer graphene with large single domains [17,18].

Since single-layer graphene is usually of higher interest, a large amount of experimental and theoretical studies were dedicated to this process with the aim to find optimal conditions. As a result, low-carbon-source conditions have been found to be preferable, and they are typically used in industrial production. Moreover, high temperature, which suppresses the formation of dendritic domains, is usually used. In this work, we concentrate on the CVD growth process on Cu(111) that uses methane as a carbon source and occurs at high temperatures (c.a. 1300 K).

Theoretical methods are widely used to support experimental findings and reveal important details of the mechanisms underlying the CVD production of graphene. From a theoretical point of view, if side-processes (like defect healing or processes related to surface imperfections) are ignored, one can distinguish the following essential steps involved in the CVD:The surface-catalyzed pyrolysis of hydrocarbons that lead to numerous carbon species on the surface;The aggregation and nucleation of these species on the metal surface, which produces graphene nuclei;The growth of the nuclei due to the attachment of feeding species;The coalescence of the flakes when high coverage is reached.

Each of these steps involve different elementary reactions, and the majority of steps occur simultaneously, influencing each other. This impacts the overall kinetics of the CVD process and, consequently, the quality of the resulting material. Due to such complexity of the process, its modeling requires multiscale approaches for a proper description.

The most popular way to study CVD within multiscale settings is to combine a kinetic Monte Carlo (kMC) approach with DFT calculations [19,20,21,22,23]. DFT is used to investigate elementary reactions at atomistic levels and to calculate corresponding energy barriers, which are then utilized by kMC in describing the kinetics of the flake growth. One of the main conclusions of these studies is that on Cu(111) monoatomic and diatomic carbon are the most important candidates, playing a key role for the dominant feeding particle of the growth process (i.e., it was shown that they have low diffusion barriers/high mobility on the surface), yet, to our knowledge, there is no general agreement on this question in the literature [24,25,26,27]. Carbon trimer was evaluated as an important feeding material on h-BN [28]; however, on copper it is believed to be much less significant [29].

Although the kMC+DFT methodology allows the simultaneous accounting for many species and reactions which can take part in the process, it is important to point out a few significant limitations of this approach:Simulations are performed in the range of up to several hundred nanometers, whereas the typical sizes of the nuclei observed in experiments are in the micrometer scale [30];kMC calculations are usually limited to the steady growth of a single graphene nucleus, so there is no information on the nucleation step included. Consequently, no size distribution or nuclei density can be obtained, which are known to be crucial characteristics of CVD graphene;Simulations on the hexagonal kMC lattice are complicated by the formation of the pentagon edge (the pairwise closing of the Klein edge of graphene on the Cu(111) surface) and its opening by the next attaching particle.

To our best knowledge, there has only been a very restricted number of attempts [30,31] to describe the nucleation and time evolution of the macro-ensemble of graphene flakes within kinetic nucleation theories, where the compact phase is formed from the 2D lattice gas of feeding particles. Therefore, in this work, we develop an analytical kinetic model of the nucleation and growth of graphene on the Cu(111) surface which, in analogy with kMC, utilises DFT barriers to express the rate constants of elementary steps. Since $C_{2}$ adparticles are known to be very stable, they could play a significant role in nuclei formation along with single C atoms. Hence, the proposed model takes into account this possibility and reformulates classical nucleation theories [32,33,34,35,36] (which usually consider nucleation from particles of the same type) to make them suitable for this case.

Moreover, we re-evaluate the DFT barriers of the most important elementary reactions and find out significant deviations from values used in recent studies. These differences qualitatively influence the results of kinetic modeling.

## 2. Theory and Computational Details

At the first stage of the CVD process, methane adsorbs and decomposes on the Cu(111) surface, producing carbon adatoms, which then agglomerate and eventually form the graphene nuclei. For our further discussion, it is convenient to divide carbon containing surface particles into three classes:C and $C_{2}$ adparticles which have high mobility on the surface and play the role of feeding species for nucleation and growth;Small $C_{n}$ clusters of various geometries which cannot yet be associated with the solid crystalline graphene phase;Well-defined graphene nuclei which are immobile on the surface and grow due to the attachments of C and $C_{2}$ particles.

We also assume that when the number of atoms in the cluster *n* reaches a certain value $n_{*}$, the cluster of the second type becomes a well-defined nucleus of the graphene phase (third type). The determination of the value $n_{*}$ for different conditions is closely related to the problem of "magic carbon clusters" [37] known in the literature and presents a significant challenge that is beyond the scope of the present paper. Moreover, a detailed study of the kinetics of small $C_{2}$ adparticles of the second type is extremely complicated due to a very substantial number of reactions and energy barriers which have to be taken into consideration. To avoid these obstacles, we assume that the kinetics of the formation of $C_{n_{*}}$ nuclei satisfy the quasistationary approximation and can be described by a nucleation rate *I* depending on the surface concentration of C and $C_{2}$ as discussed below.

As it may be evidenced from various microscopic images [30,38,39,40,41,42], under high temperatures, growing graphene nuclei tend to have well-defined hexagonal or circular shapes. In this work, we consider graphene nuclei shaped in the forms of perfect hexagons; however, generalization to the case of circular shape is straightforward and does not significantly affect the formulation of the theory. This approximation allows us to connect the radius of the flake *r* with the number of atoms *n* in it:(1)$$n=\frac{3\sqrt{3}}{2}\rho_{s}r^{2}$$
where $\rho_{s}=3.92\cdot 10^{19}\;\left(m^{-2}\right)$ is the density of carbon atoms in the graphene lattice.

In addition, we assume that the edges of the growing nuclei have the most stable zigzag form, so the attachment/detachment processes may be characterized by the respective barriers calculated with DFT for the zigzag edge as described below.

Within all introduced approximations, we consider the following kinetic scheme (see Figure 1 for a pictorial representation):$${\mathrm{CH}}_{4(g)}\overset{k_{1}}{\to }C^{*}+2H_{2(g)}$$
$$C^{*}+C^{*}\overset{k_{d}}{⇌}C_{2}^{*}$$
$$C^{*}+{\mathrm{nucleus}}^{*}\overset{k_{at}\left(C\right)}{\underset{k_{det}\left(C\right)}{⇌}}{\mathrm{nucleus}}^{*}$$
$$C_{2}^{*}+{\mathrm{nucleus}}^{*}\overset{k_{at}\left(C_{2}\right)}{\underset{k_{det}\left(C_{2}\right)}{⇌}}{\mathrm{nucleus}}^{*}$$
where species marked with * are adsorbed on the surface and the meaning of the rate constants $k_{i}$ is explained further. The expressions of the rate constants through respective energy barriers may be found in the Appendix B. It is important to note that, in the proposed model, we neglect the impact of $C_{x}H_{y}$ particles on the nucleation and growth kinetics and apply the quasistationary approximation for the surface concentration of the intermediate products of the step-wise (and surface-catalyzed) decomposition of methane. This allows us to express the rate of C adatoms production *J* as the rate of the dissociative adsorption of ${\mathrm{CH}}_{4}$ (see Appendix A for more details).

We also note that the model is derived for the case of low surface coverage (*c.a.* <10%) so that the whole surface area is assumed to be available for the dissociative adsorption of methane.

While in some papers [30,43] the desorption of a carbon monomer is taken into consideration, we evaluate its binding energy as 4.98 eV (PBE-D3) and its lifetime by orders of magnitude higher than the considered timeline. Thus, the process of re-evaporation is considered as insignificant.

### 2.1. Growth Rate

The growth rate of an individual graphene nucleus of hexagonal shape consisting of *n* atoms has the following form:(2)$$v=\frac{dn}{dt}=r_{at}^{C}\left(n\right)-r_{det}^{C}\left(n\right)+2r_{at}^{C_{2}}\left(n\right)-2r_{det}^{C_{2}}\left(n\right)$$
where $r_{at}^{i}\left(n\right)$ and $r_{det}^{i}\left(n\right)$ are the rates of attachment/detachment of the *i*-th adparticle (C or $C_{2}$) to/from the nucleus. Taking into account the relation in Equation (Equation 1) between *n* and the flake’s perimeter (P=6r) and applying a 2D lattice gas model for adparticles, one can express these rates as
(3)$$r_{at}^{i}\left(n\right)=k_{at}\left(i\right)c\left(i\right)\sqrt{n}$$
(4)$$r_{det}^{i}\left(n\right)=k_{det}\left(i\right)\sqrt{n}$$
where $c\left(i\right)$ is the surface density (concentration) of the *i*-th particle and $k_{at}\left(i\right)$ and $k_{det}\left(i\right)$ are attachment/detachment rate constants which can be calculated from the corresponding barriers given in Table 1 as explained in the Appendix B.

Applicability of the 2D lattice gas model is clarified in Appendix A.

It is convenient to introduce equilibrium surface concentrations $c_{eq}\left(i\right)$ corresponding to the dynamic equilibrium ($r_{at}^{i}=r_{det}^{i}$) between the solid graphene phase and the 2D lattice gas consisting of *i*-th particles *only*
(5)$$c_{eq}\left(i\right)=\frac{k_{det}\left(i\right)}{k_{at}\left(i\right)}$$
and the respective oversaturation parameters
(6)$$\xi_{i}=\frac{c\left(i\right)-c_{eq}\left(i\right)}{c_{eq}\left(i\right)}$$.

Within such notations, one can express the growth rate *v* as
(7)$$v\left(n,t\right)=\frac{2}{t_{1}}\left\{\xi_{C}\left(t\right)+\frac{2t_{1}}{t_{2}}\xi_{C_{2}}\left(t\right)\right\}\sqrt{n}=\frac{2}{t_{1}}\xi \left(t\right)\sqrt{n}$$
where $t_{1}$ and $t_{2}$ are defined as follows:$$t_{1}=\frac{2}{k_{at}\left(C\right)c_{eq}\left(C\right)}$$
$$t_{2}=\frac{2}{k_{at}\left(C_{2}\right)c_{eq}\left(C_{2}\right)}$$

If $\xi \left(t\right)>0$, then the solid phase tends to grow, whereas it decays into an ensemble of adparticles when $\xi \left(t\right)<0$. Therefore, $\xi \left(t\right)$ plays the role of an effective oversaturation.

### 2.2. Kinetic Model of Growth

When $\xi \left(t\right)<0$, the kinetics of the process are described by the following equations:(8)$$\frac{d\xi_{C}}{dt}=\frac{J}{c_{eq}\left(C\right)}-2k_{d}c_{eq}\left(C\right){\left(1+\xi_{C}\right)}^{2}$$
(9)$$\frac{d\xi_{C_{2}}}{dt}=\frac{k_{d}c_{eq}^{2}\left(C\right)}{c_{eq}\left(C_{2}\right)}{\left(1+\xi_{C}\right)}^{2}$$
from which one immediately obtains
(10)$$\xi_{C}\left(t\right)=\sqrt{\frac{J}{2k_{d}c_{eq}^{2}\left(C\right)}}\tanh\left[\sqrt{2Jk_{d}}t\right]-1$$
(11)$$\xi_{C_{2}}\left(t\right)=\frac{J}{2c_{eq}\left(C_{2}\right)}\int_{0}^{t}\tanh^{2}\left[\sqrt{2Jk_{d}}t^{\prime }\right]dt^{\prime }-1$$

The duration τ of the stage when the solid phase is unstable against the lattice gas is defined by the following equation:$$\xi_{C}\left(\tau \right)+\frac{2t_{1}}{t_{2}}\xi_{C_{2}}\left(\tau \right)=0$$

Once the effective oversaturation $\xi \left(t\right)$ reaches positive values, an ensemble of C and $C_{2}$ adparticles becomes unstable against the solid phase. Starting from this moment, two competing processes must be considered: (i) the formation of new nuclei with $n_{*}$ carbon atoms and (ii) the growth of existing nuclei. The relative dynamics of these processes define the time evolution of the monomer and dimer concentrations and the resulting size distribution of graphene flakes on the surface. The latter is characterized by a function $g\left(n,t\right)$ describing the distribution of nuclei according to the number of carbon atoms *n* in them for a given moment of time *t*. This function satisfies the continuity relation, manifesting itself in the form of the Fokker–Planck equation
(12)$$\frac{\partial g\left(n,t\right)}{\partial t}+\frac{\partial }{\partial n}\left(v\left(n,t\right)g\left(n,t\right)\right)=0$$
with the following boundary and initial conditions:(13)$$g\left(n_{*},t\right)=\frac{I\left(\xi \left(t\right)\right)}{v\left(n_{*},t\right)}$$
(14)$$g\left(n,0\right)=0$$

In these equations, *I* is the rate of the formation of nuclei with $n_{*}$ atoms (nucleation rate) and *v* is the growth rate introduced in the previous subsection.

Solving Equations (Equation 12)–(Equation 14) together with the mass balance equations for C and $C_{2}$ allows for obtaining the concentration profiles of adparticles and the size distributions of graphene flakes formed during the CVD process. A detailed description of the methodology used to solve such kinetic equations for any functional forms of $I\left(\xi \left(t\right)\right)$ and $v\left(n,t\right)$ is given in Ref. [33]. Therefore, we only give a brief sketch of the mathematical technique applied to obtain the results of the present work (see Appendix A).

### 2.3. Nucleation Rate

To solve the system of kinetic equations described in the previous subsection, one has to define the rate of nuclei formation *I* as a function of the effective oversaturation $\xi \left(t\right)$ and the relative concentrations of C and $C_{2}$ in the system. In principle, to derive the nucleation rate in this case, one has to perform a detailed modeling of the kinetic behavior of small $C_{n}$ clusters using ab initio MD or kinetic Monte Carlo simulations based on the rates calculated by DFT or high-level quantum chemical methods. This task, however, presents a significant challenge from the computational point of view and goes beyond the scope of this work. At the same time, applying classical nucleation theories (such as the Lifshitz–Slyozov [34,35] or Becker–Döring [36] theories) originally developed for nucleation driven only by monomer attachment does not seem to be straightforward for the case of graphene nucleation, where dimer attachment plays a significant role.

To avoid these obstacles, we introduce a semiempirical nucleation rate, in which functional dependence on the effective oversaturation $\xi \left(t\right)$ mimics the one derived within the Lifshits–Slyozov theory for the disc-shaped surface nuclei:(15)$$I\left(\xi \left(t\right)\right)=A\left(\xi +1\right)\sqrt{\ln\left(\xi +1\right)}\exp\left(-\frac{U}{\ln\left(\xi +1\right)}\right)$$
while parameters *A* and *U* are extracted from experimental data as discussed in Section 3.2.

Strictly speaking, the parameter *U* defining a barrier to nuclei formation must depend on the relative concentrations of monomers and dimers; however, in the present contribution, we consider it as an effective parameter which remains constant throughout the process.

It is important to note that the nucleation mechanism considered here cannot be described in terms characteristic for theories applied to multicomponent nucleation. This is due to the fact that C and $C_{2}$ can be formed from another, whereas the chemical potential in bicomponent nucleation is a sum of the chemical potentials of both components.

### 2.4. Details of DFT Calculations

Elementary reactions and reaction rates have been calculated at the DFT level as follows: We employed the BEEF-vdW [44] and PBE [45,46,47,48]-D3 [49] functionals using the Vienna ab initio simulation package (VASP 5.4) [50,51]. The differences in using the D3 vs. the D2 dispersion correction have been evaluated by comparison with data in the literature. While studies in previous works used a rather small simulation cell [19] (Figure 2b), we modeled the Cu(111) surface by a (10 × 3) slab and increased the width of the graphene ribbon to a "five-rings-wide" ribbon (Figure 2a) in order to eliminate the short-range effects originating from too thin ribbons and to keep a significant inter-ribbon distance, which we identified to be crucially important for the proper calculation of reaction barriers (see the discussion of DFT in the next section). To ensure that there is no interaction between periodic images, these were separated by a vacuum region of at least 12 Å perpendicular to the surface. All calculations were conducted according to the non-spin-polarized scheme. More details of the DFT calculations can be found in the Appendix A.

## 3. Results and Discussion

### 3.1. DFT Calculations

Despite the fact that there are just a few reaction barriers required in our model and that those were previously reported by Li et al. [19], our re-evaluation of the important calculations showed drastic differences in the most important attachment barriers, namely the attachment of C and $C_{2}$ to the zigzag edge. In particular, the attachment barrier of $C_{2}$ was reported to be 0.58 eV in the cell shown on Figure 2b using the PBE-D2 functional (Appendix A), while our results were 1.25 eV using the PBE-D3 functional. The difference of 0.6 eV is too large to be caused by differences between D2 and D3 and, indeed, using the same small cell, we managed to reproduce a similar value to that reported by Li with PBE-D3 and BEEF-vdW functionals. In contrast, the detachment barriers were in good agreement. Therefore, we investigated whether the "two-rings-wide" ribbon could be used here. We could expect that, for larger graphene particles, electronic and geometric effects might change significantly. Therefore, we recalculated the barrier for the attachment/detachment of $C_{2}$ to graphene ribbons of different widths (Table 2) using the 10 × 3 copper slab as explained above.

Table 2 indicates that the effect of ribbon width is insufficient to explain the differences with the data reported by Li, but we found that the reason lies in the too small inter-ribbon gap. Indeed, the adsorption of $C_{2}$ in the gap of the literature model is 0.6 eV less favorable than on the clean copper surface. Despite the fact that the coadsorption effect is strongly suppressed in our model (as one can see from Table 2), we calculated the attachment barriers in this work with respect to individualized particles. With this approach, we repeated the calculations of Li et al. with the PBE-D2 functional and obtained a barrier of 1.25 eV for the attachment of $C_{2}$ to the graphene zigzag edge (Figure 3), which is in excellent agreement with our results.

In Table 1, the results for all reaction barriers are compared. Compared to Li et al., there are two major differences in the reaction barriers: not only the attachment of $C_{2}$ to graphene, where our barrier is much larger as already explained above, but also the attachment of C to the graphene zigzag edges is substantially different. While Li et al. reported a barrier of 1.27 eV for the attachment of carbon, we found a much lower barrier (0.57 eV with PBE-D3 and 0.68 eV with BEEF-vdW).

Overall, our DFT results show that the barriers of the two crucially important concurrent reactions, the attachment of C and $C_{2}$ to the graphene zigzag edges, are drastically changed in opposite directions in comparison to the literature data. Taking into account that barriers enter the reaction rates exponentially, the role of the feeding species in graphene growth needs to be revised.

### 3.2. Analytical Kinetic Model

In this section, we analyze the solution of the kinetic system from Section 2.2 with the values of kinetic parameters calculated using BEEF-vdW barriers and collected in Table 3. Parameters *A* and *U* entering Equation (Equation 15) were fitted in a way that the solution for the size distribution function $g\left(n,t\right)$ reproduces experimentally observed distribution widths reported in Ref. [30], where the CVD process was performed in conditions very similar to the ones considered here. We also note that, according to our analysis, kinetic parameters calculated with PBE-D3 barriers yield qualitatively the same results, so the main conclusions discussed below remain valid for this set of barriers as well. Our choice of the BEEF-vdW set of barriers is due to the fact that this functional is known to provide better results for the energetics of the surface processes [52,53,54,55].

The effective oversaturation $\xi \left(t\right)$ and concentration profile of $C_{2}$ are plotted in Figure 4a,c. The profile of carbon adatoms (see Appendix A) demonstrates the trivial behaviour characteristic of highly reactive intermediates—its concentration quickly (in $2\times 10^{-5}$ s) reaches its small asymptotic value ($∼\sqrt{J/2k_{d}}=3.67\times 10^{12}\;\left(m^{-2}\right)$) and does not change significantly throughout the process, so the quasistationary approximation holds ($dc\left(C\right)/dt\approx 0$). This happens due to the fact that the barrier of dimer formation is significantly lower compared to the barriers of all other reactions included, so the fast dimerization rate does not allow adatoms to accumulate in the system. Due to the quasistationary behavior of C concentration (Appendix A), the functional shape of the effective oversaturation and $C_{2}$ concentration profile should be the same, as can be evidenced from Figure 4a,c.

Analyzing the described concentration profiles allows us to identify the roles of the monomer and dimer in both nucleation and growth processes. First of all, it is interesting to point out that the steady-state concentration of C corresponds to a substantially negative value of $\xi_{C}\approx -0.99$, meaning that at any moment of time throughout the process graphene flakes are not stable against this population of carbon adatoms. This particular force drives nuclei to decay, releasing single atoms. Simultaneously, as it follows from Figure 4a, once the effective oversaturation $\xi \left(t\right)$ becomes positive, it stays that way throughout the whole process, slowly approaching its asymptotic positive value over time. This indicates that, despite decay against single atoms, graphene flakes eventually grow due to the dimer attachments prevailing in the system. This fact may seem surprising if one simply compares barriers of C and $C_{2}$ attachments (0.57 eV vs. 1.21 eV), concluding that carbon adatoms must attach easier and prevail in the growth mechanism. In fact, the observed kinetic regime is determined by a combination (or competition) of two factors: (i) the tremendous stability of $C_{2}$ compared to single atoms (hence, a high rate of dimerization) and (ii) the higher rate of attachment of single atoms compared to that of dimers. In our case, the first factor prevails, allowing us to accumulate dimers in a concentration big enough to make $C_{2}$ responsible for the growth process. Furthermore, it keeps the C concentration low enough such that the flakes decay against the adatoms population. Therefore, we observe an interesting kinetic regime illustrated in the Graphical Abstract: graphene nuclei, being unstable against monomers, release single atoms; released single atoms quickly dimerize on the surface and incorporate back into graphene flakes in the form of dimers, making $C_{2}$ attachment the main driving force for graphene growth. The same conclusion is valid for the nucleation process: as we can see from Figure 4a–c, the sharp peak of the effective oversaturation observed between 0.95 and 1.1 seconds and responsible for active nucleation in the system is attributed to the peak on the $C_{2}$ concentration profile. The observed central role of $C_{2}$ adparticles in the growth process was previously reported in Ref. [29], where the growth process was studied by kMC. We believe that our analysis, on the one hand, strengthens this hypothesis and, on the other, deepens our understanding of the respective roles of C and $C_{2}$ in both the growth and nucleation processes. The latter is particularly interesting, as it shows that graphene nucleation follows an unusual mechanism, where the dimer oversaturation is responsible for nuclei formation, while the monomer effectively increases the barrier of nucleation (because $\xi_{C}<0$). This has not been taken into account in the analysis of graphene nucleation patterns reported before [30], and further investigations of such mechanisms could potentially reveal interesting details of graphene nuclei formation in surface deposition processes. It is also important to note that the model is developed for low surface coverages and its conclusions have to be extrapolated to the later stages of the growth process with a certain caution. For example, one can speculate that at the late stages where the available Cu(111) surface area is low, the dimerization process could get suppressed and the role of monomers will be different. However, the coalescence of the particles and the deviation of their shapes from the hexagonal ones [40] at the later stages might make the proposed model inapplicable here.

Another important quantity which follows from the solution of the kinetic model is the size distribution function $g\left(n,t\right)$ shown in Figure 4d for the moment of time t=1.5s. We note that the shape of this function would be the same for all further moments, since no additional nucleation will appear (as $\xi \left(t\right)$ reaches its steady state) and the growth rate of the flake’s linear size does not depend on the number of atoms in the flake, as can be seen from Equation (Equation 7). Therefore, the observed width of the distribution function is a characteristic value and it was used to fit parameters *A* and *U* from the experimental data [30] mentioned above.

Finally, we compare the calculated values of nuclei’s total density and mean growth rate ($4.95\times 10^{8}\;\left(m^{-2}\right)$ and $291\;\left(\mu m^{-2}\cdot s^{-1}\right)$, respectively) with experimental data [30]. As can be seen, the calculated values deviate significantly from the experiment: the mean growth rate is two orders of magnitude higher while nuclei density is two orders of magnitude lower. According to our analysis, this cannot be fixed by any reasonable reparameterization of the nucleation rate parameters (*A* and *U*), and the main reason for this substantial difference lies in the high values of the attachment rate constants ($k_{at}$) for both monomer and dimer. Our analysis shows that, for example, increasing the corresponding BEEF-vdW barriers ($E_{at}$ and $E_{det}$) simultaneously by 0.20–0.30 eV allows us to perfectly reproduce the experimental values of the mean growth rate and nuclei density. We note that such a deviation does not seem to be very significant and lies within the range of typical DFT errors [52,53]. As our discussion from the previous subsection demonstrates, energy barriers are quite sensitive to the used functional and chosen methodology of calculations. However, even such relatively small changes in barriers can significantly impact the numerical results of the kinetic model, since the barriers enter the equations for the rate constants exponentially. We must also note that the qualitative conclusions about nucleation and growth mechanisms made above are much less sensitive to the chosen barriers and remain valid for the shifted values as well.

## 4. Conclusions

We developed an analytical kinetic model of the nucleation and growth of graphene on Cu(111) that simultaneously includes C and $C_{2}$. Applying DFT barriers for the calculation of reaction constants and fitting two parameters (*A* and *U*) to experimental data, we were able to construct a consistent description of the process. Among the most important results, the role of C has to be reconsidered, since we observed the instabilities of the graphene flakes with respect to the adatoms population at any point of time. With extremely low computational cost, this model allows us to analyze the dependencies of the basic parameters (the time and length of nucleation, size distribution and total nuclei density as functions of time, etc.) from input parameters: temperature, the pressure of feeding gas, and reaction barriers on a given surface. It is important to note that, being tested for the CVD process with the thermal splitting of methane on the Cu(111) surface, the model can be easily (by recalculating barriers) applied to other metal or alloy surfaces with low carbon solubility, other feeding gases, and even to plasma-enhanced CVD.

## Figures and Tables

**Figure 1 nanomaterials-12-02963-f001:**
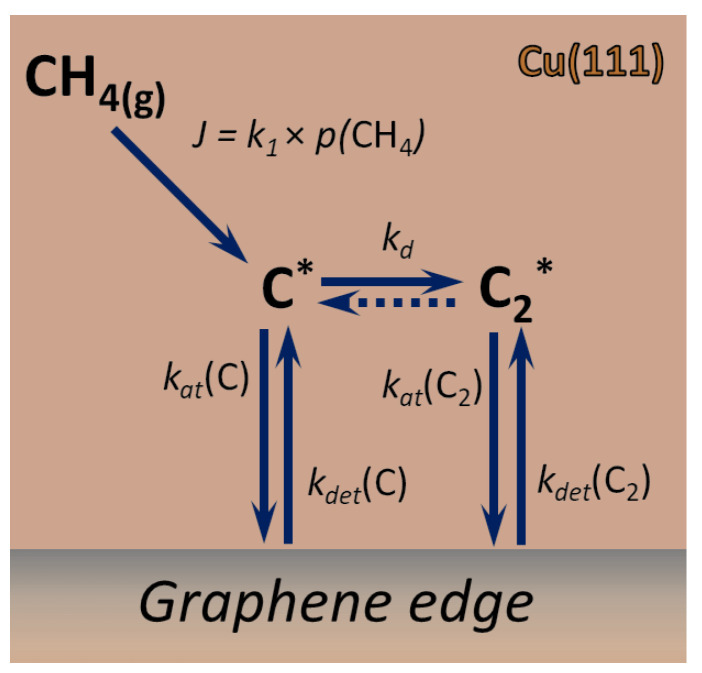
Schematic illustration of the model discussed in the paper. Top view on the surface of Cu(111) with graphene edge. ${\mathrm{CH}}_{4(g)}$ corresponds to gas-phase methane, while all species labeled with * are adsorbed on the surface. Methane dissociation and all following processes are taking place on the surface and are surface-catalyzed.

**Figure 2 nanomaterials-12-02963-f002:**
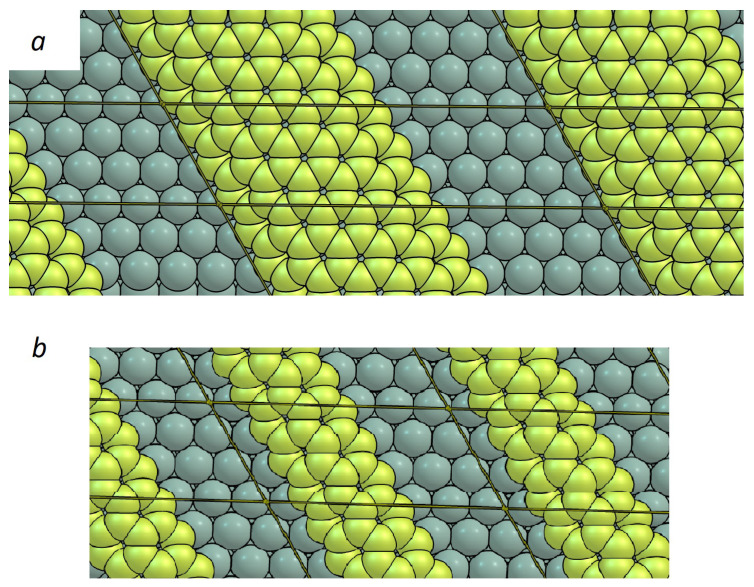
(**a**) Depiction of graphene ribbons in the simulation cell used in this work. (**b**) Depiction of the smaller model used in [19].

**Figure 3 nanomaterials-12-02963-f003:**
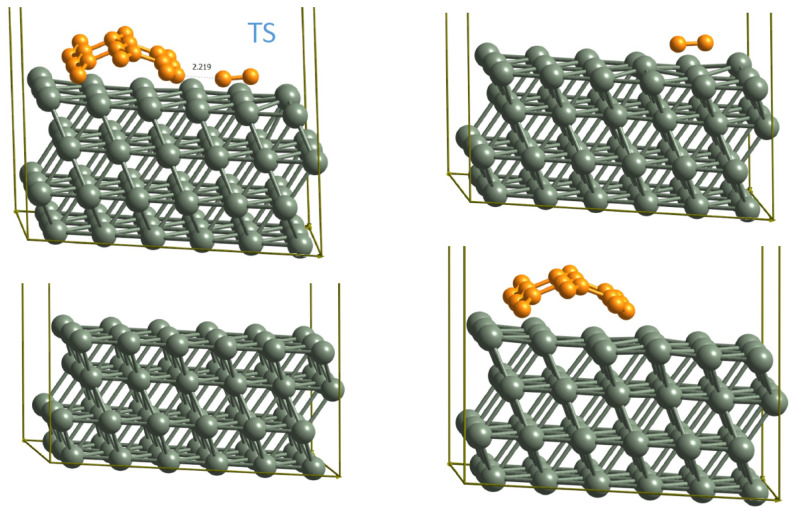
Attachment barrier of $C_{2}$ to the graphene zigzag edges on the Cu(111) surface calculated with respect to two individualized species. All structures involved are shown. The barrier is calculated in the following manner: $E_{\mathrm{Barrier}}=E_{\mathrm{TS}}-E_{C_{2}}-E_{\mathrm{ribbon}}+E_{\mathrm{surface}}$.

**Figure 4 nanomaterials-12-02963-f004:**
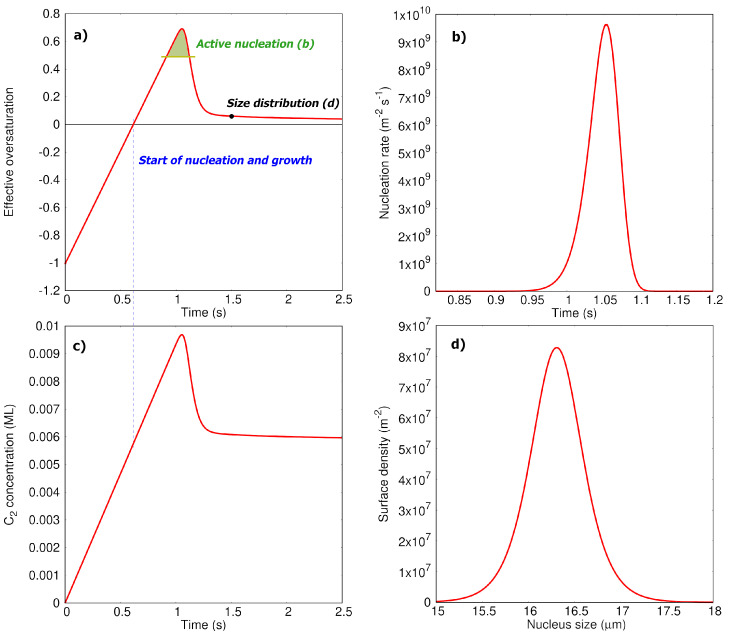
(**a**) Effective oversaturation, which depends simultaneously on oversaturation/ concentration of C and $C_{2}$ (see Equation (Equation 7)), as a function of time. It reaches zero at 0.615 seconds, which corresponds to the duration of the stage when nucleation is not possible. Starting from 0.615 seconds, the solid phase becomes stable against the lattice gas of adparticles. (**b**) Nucleation rate as a function of time. Note different time scales—the major amount of nuclei are formed in a short period of time—from 0.95 seconds to 1.1 seconds, when the effective oversaturation reaches its maximum values. (**c**) $C_{2}$ concentration profile. One can see that the effective oversaturation reaches zero, when $C_{2}$ concentration is orders of magnitude higher than the equilibrium one $c_{eq}\left(C_{2}\right)$ (which is around $5\times 10^{-5}$ ML (monolayer), Appendix A). (**d**) Size distribution of graphene flakes built at 1.5 seconds. Note that the shape and width of this curve will not change after the end of active nucleation; just the absolute size of the particles will increase.

**Table 1 nanomaterials-12-02963-t001:** Reaction barriers in eV.

Reaction	BEEF-vdW	PBE-D3	Ref. [19]
${\mathrm{CH}}_{4(g)}\to C^{*}+2{H_{2(g)}}^{a}$	1.94	1.48	1.63
$C^{*}+C^{*}\to C_{2}^{*}$	0.25	0.55	0.25
$C_{2}^{*}\to C^{*}+C^{*}$	3.97	3.62	2.75
$C^{*}+\mathrm{edge}\to \mathrm{edge}-C$	0.57	0.68	1.27
$\mathrm{edge}-C\to C^{*}+\mathrm{edge}$	1.76	1.36	1.57
$C_{2}^{*}+\mathrm{edge}\to \mathrm{edge}-C_{2}$	1.21	1.32	0.58
$\mathrm{edge}-C_{2}\to C_{2}^{*}+\mathrm{edge}$	2.31	2.30	2.19

^a^ Barriers are given for the reaction ${\mathrm{CH}}_{4(g)}\to {\mathrm{CH}}_{3}^{*}+H^{*}$ because it determines the $C^{*}$ production rate *J*. All energies in the table are given as enthalpies. Graphical representation of Gibbs free energies on the path ${\mathrm{CH}}_{4}^{*}\to C^{*}+2H_{2(g)}$ can be found in Appendix A.

**Table 2 nanomaterials-12-02963-t002:** $C_{2}$ attachment/detachment barriers in eV for different ribbon widths calculated using the BEEF-vdW functional.

Ribbon Width	Attachment Barrier (eV)	Detachment Barrier (eV)
two-ring	1.21	2.40
three-ring	1.32	2.38
four-ring	1.25	2.28
five-ring	1.21	2.31

**Table 3 nanomaterials-12-02963-t003:** Values of kinetic parameters calculated with BEEF-vdW barriers for T=1300K and $p\left({\mathrm{CH}}_{4}\right)=10\;\mathrm{Torr}$. Parameters *A* and *U* are fitted from experimental data on size distribution Ref. [30].

Kinetic Parameter	Units	Value	Physical Meaning
*J*	$\left(m^{-2}\cdot s^{-1}\right)$	$7.32\times 10^{17}$	Rate of C production
$k_{d}$	$\left(m^{2}\cdot s^{-1}\right)$	$2.71\times 10^{-8}$	Rate constant of C dimerization
$k_{at}\left(C\right)$	$\left(m^{2}\cdot s^{-1}\right)$	$6.94\times 10^{-10}$	Rate constant of C attachment to a flake
$k_{det}\left(C\right)$	$\left(s^{-1}\right)$	$7.26\times 10^{5}$	Rate constant of C detachment from a flake
$k_{at}\left(C_{2}\right)$	$\left(m^{2}\cdot s^{-1}\right)$	$1.62\times 10^{-12}$	Rate constant of $C_{2}$ attachment to a flake
$k_{det}\left(C_{2}\right)$	$\left(s^{-1}\right)$	$3.09\times 10^{3}$	Rate constant of $C_{2}$ detachment from a flake
*A*	$\left(m^{-2}\cdot s^{-1}\right)$	$2.0\times 10^{22}$	Rate of nucleation assuming zero nucleation barrier
*U*	-	15	Value of U/ln(ξ+1) - defines nucleation barrier for the given effective oversaturation ξ

## Data Availability

Not applicable.

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
