# Peer review of "Analytical Model of CVD Growth of Graphene on Cu(111) Surface"

_nanomaterials, 2022, doi:10.3390/nano12172963_

Round 1

Reviewer 1 Report

The manuscript entitled “Analytical model of CVD growth of graphene on Cu (111) surface” established a theoretical model to describe the graphene nucleation and growth on Cu (111) surface based on the conbination of kinetic nucleation theory and DFT simulations. Such model enables the authors to analyze the dependencies of the basic parameters (time and length of nucleation, size distribution and total nuclei density as functions of time, etc.) from input parameters: temperature, pressure of feeding gas, reaction barriers on a given surface. The whole manuscript is well written and fits well with the scope of the journal as well as the topic of the special issues. Therefore, I recommend MINOR revision before it can be published. The following questions are listed for the authors’ reference:

(1) The authors believed that the transformation from C to C2 correspond to the maximum nucleation rate (as shown in Figure 4a and c). Are there any experimental results in the relevant literature that can confirm the above conclusion?

(2) If the transformation from C to C2 do exists, why the concentration of C almost not change in the general timescale of the process? (Figure S3)

(3) The authors should provide a simple intuitive schematic image to represent the adsorption, transformation of C atoms as well as the final nuclei formation process.

Author Response

Please find answers in attached file.

Reviewer 2 Report

The presented growth theory (actually classical) of graphene is clear and mathematically understandable. It could also be applied to the vapor phase growth of other carbon materials such as diamond for example. I miss some reference to the real, experimental (e.g. SEM photos) growth steps of this material. It would be a much more valuable work.

Line 46:

Does nucleation take place in the gas phase or on the surface of the substrate? This must be clearly stated. Do substrate surface defects play a role?

Line 53: Please expand the abbreviations in the text

Author Response

Please find answers in attached file

Reviewer 3 Report

The manuscript “Analytical model of CVD growth of graphene on Cu (111) surface" describes an interesting kinetic regime that might occur under certain conditions. The authors propose to broaden the attention to carbon intermediates governing the process. I believe both the discussion and results to be of interest to the community of the Nanomaterials. Nevertheless, description of a few findings should be improved to remove ambiguity for the broader audience.

The following comments should be addressed for the manuscript to be suitable for further discussion.

1.       Methane decomposition pattern is one of the biggest concerns.

1.1.    Why the authors solely consider the gas phase non-catalytic pyrolysis as the only channel producing intermediate carbon species? Catalyst surface as a rule significantly decreases energy barriers for hydrocarbon decomposition. In this light, the discrepancy discussed in the last paragraph l. 304-307 might occur from that point.

1.2.    Under high vacuum, such energetically greedy first-order (as the authors present it) reactions can be also limited by energy relaxation within ensemble (e.g. Lindemann mechanism or RRKM theory).

1.3.    Moreover, the methane decomposition might occur not only on Cu surface but also in vicinity of graphene perimeter (10.1002/advs.202200217) which might also alter the energetic path.

1.4.    Last but not least, the paper sometimes (may be unintentionally) regards methane decomposition as an elementary reaction (A→B goes for dA/dt=-kA), which is apparently not (fig. S2). However, at least in figure 1 and appendix A the reaction is regarded in such a way. The authors should definitely clarify why such an approach is applicable here.

2.       Intermediates:

2.1.    Authors should provide more information on the experimental observations in literature on ratio between C and C2 as here more than three orders of magnitude of difference is shown.

2.2.    IF dimers are so stable, why not to consider trimers (C3*)? The possible finding that their concentration would be lower should give one more unambiguous assessment of the model proposed.

2.3.    The fact that the C+C=C2 reaction is faster then C+nucleus=nucleus is very interesting. Intuitively, the mean free path of C should be a function of surface coverage of C and some specific  perimeter density for graphene edges. Thus under some conditions, e.g. near fully covered surface the direct accumulation of C should prevail. Would the authors be so kind to discuss on where is that boundary as C concentration is 10^-7 ML only and second order reaction between such species should be rare?

3.       Kinetic model:

3.1.                         L. 355 Why v0 is of 10^12 s-1 here? To my knowledge this is derived from kb*T/h (or Eyring-Evans-Polyani theory) which is under that conditions studied (1300 K should be 2.7e13

3.2.                         Apart from concerns on methane decomposition and A1, authors should significantly enlarge discussion (may be in SI) on how such formulas (A2-A4) were derived. I would like to remind that for 2D processes on the surface the partition functions used of transition state theory are different from those for usual gas reactions in 3D.

3.3.    In papers 10.1021/nn3008965 and 10.1002/advs.202200217, C desorption is considered as one of the reactions playing significant role in the nucleation pattern at high temperatures. What is the reason for taking it out of consideration here?

3.4.    Though direct origin of kinetic constants in not clear, it appears for me that the diffusion of C and C2 was not taken into account (once again unlike 10.1021/nn3008965 and 10.1002/advs.202200217). Nevertheless, the diffusion rate might be comparable to the reactions considered and play a significant role. What is the reason for that?

Minor comments:

·        - L. 53. Though the abbreviations are given in the end, it would really help if authors would introduce abbreviations at the points of first appearance (like KMC in this case)

·         -Table 1. I advice you to add units for energies within the table somewhere to remove ambiguity.

·         -Figure 1 is a bit misleading. Apart from the abovementioned concerns related to methane decomposition, the space arrangement is not clear. Is it top view on the surface of Cu (111) with graphene edge? Then why CH4 is on the surface? If it is a side view, why methane is within copper? Actually, given that methane pyrolysis is solely gas phase here where recombination C to C2 happens? It is not clear as well from Figure 1.

·        - L. 104: the quasi-stationary approach is proposed for Cn* which is formation of nuclei. Authors do not prove this approach by any of criteria (unlike the case of C* l. 251-260). Please clarify.

·         -Line 118. Just out of scientific curiosity... Would the authors be so kind to discuss (may be in SI) the equilibrium concentrations of C and C2 for reactions #3 and #4 at the line 118?

·        - Page 4: you use three designations (j, r, v) for the reaction rate which really hinders the flow. What is the reason for this? How these three differ?

·         -Fig. 4d. + l. 296: Authors discuss that the distribution function for grains might be fingerprint without any further advancement. Is it possible to compare the data  with literature here?

·         -Once again, just out of curiosity, is it possible to compare C and C2 channels? E.g. estimate (may be in SI) what fraction of graphene comes from C +nucleus reaction and what from C2 + nucleus?

·        - I strongly encourage you to put figures near the places of first mention, not before or two pages after. It really hurts.

The updated manuscript should be reviewed at least one more time.

Author Response

Please find answers in attached file.

Round 2

Reviewer 3 Report

I thank the authors for explicit answers to most of the comments. Here are some minor issues remaining:

·         I propose to authors to consider explicitly state that the proposed method describes only the first stage or early moments of graphene growth as “we have to point out, that in the present paper we consider the stage of the growth process corresponding to the low occupancy of the surface by graphene flakes (< 10%)”. I am still failing to observe this statement anywhere in the text. From the abstract/conclusions, it seems like all the growth pattern was considered.

·         “We show that calculated diffusion barriers are significantly lower than attachments barriers of the same particles (i.e., the highest diffusion barrier - of dimer, 0.48 eV, is more than three times lower than the attachment barrier of C2). Thus, based on our DFT calculations, diffusion should be a minor process, so C and C2 adparticles can be treated within the lattice-gas model, where the concentration equalization happens in negligible time”. I am a bit confused. So you observe the energy barrier for diffusion which is significantly larger kT but still consider a 2D gas. Maybe this is the reason of “the mean growth rate is 320 by two orders of magnitude higher” (l.320).

Author Response

Additional English editing was performed on manuscript.
Please find answers to second round in attached file. 
